# Gender Transformative Interventions for Perinatal Mental Health in Low and Middle Income Countries—A Scoping Review

**DOI:** 10.3390/ijerph191912357

**Published:** 2022-09-28

**Authors:** Archana Raghavan, Veena A. Satyanarayana, Jane Fisher, Sundarnag Ganjekar, Monica Shrivastav, Sarita Anand, Vani Sethi, Prabha S. Chandra

**Affiliations:** 1National Institute of Mental Health and Neurosciences (NIMHANS), Bengaluru 530068, India; 2School of Public Health and Preventive Medicine, University of Monash, Melbourne 3800, Australia; 3ROSHNI-Centre of Women Collectives led Social Action, Lady Irwin College, New Delhi 110001, India; 4United Nations Children’s Fund (UNICEF) Regional Office for South Asia, Kathmandu 44600, Nepal

**Keywords:** perinatal mental health, gender transformative interventions, scoping review, LMIC

## Abstract

Perinatal mental health problems are linked to poor outcomes for mothers, babies and families. In the context of Low and Middle Income Countries (LMIC), a leading risk factor is gender disparity. Addressing gender disparity, by involving fathers, mothers in law and other family members can significantly improve perinatal and maternal healthcare, including risk factors for poor perinatal mental health such as domestic violence and poor social support. This highlights the need to develop and implement gender-transformative (GT) interventions that seek to engage with men and reduce or overcome gender-based constraints. This scoping review aimed to highlight existing gender transformative interventions from LMIC that specifically aimed to address perinatal mental health (partner violence, anxiety or depression and partner support) and identify components of the intervention that were found to be useful and acceptable. This review follows the five-stage Arksey and O’Malley framework and the Preferred Reporting Items for Systematic Reviews and Meta-Analyses extension for Scoping Reviews (PRISMA-ScR) checklist. Six papers that met the inclusion criteria were included in the review (four from Africa and two from Asia). Common components of gender transformative interventions across studies included couple-based interventions and discussion groups. Gender inequity and related factors are a strong risk for poor perinatal mental health and the dearth of studies highlights the strong need for better evidence of GT interventions in this area.

## 1. Introduction

Research and policies related to perinatal mental health have demonstrated how poor mental health both in pregnancy and postpartum is prevalent in the form of anxiety and depression, and may influence pregnancy outcomes and the health of the foetus and infant [1,2]. Untreated depression during pregnancy is also associated with a risk for suicide especially in those with a severe problem or when there is associated partner violence [2,3,4]. When studied through a socio-cultural context, women generally have reported high levels of anxiety, depression and higher levels of trauma during pregnancy as well as the postpartum period [5,6,7]. Rates of anxiety and depression in pregnancy from Low and Middle Income Countries (LMIC) range between 9–65% [8], indicating the importance of addressing mental health outcomes during pregnancy as well as post-pregnancy in the region [9].

Some of the well-established risk factors for poor perinatal mental health, particularly anxiety and depression, are related to gender inequity, especially in LMIC settings. These include partner violence (Intimate partner violence (IPV), Domestic violence (DV) and Gender-based Violence (GBV)), younger age, poor social support, low education and male infant preference [10,11,12]. Other gender-based risk factors include low autonomy and decision-making power, lack of control over resources, low education and poverty [12], power dynamics within family [13], gender-based stigma and discrimination and inequitable spousal relationships [14,15]. Multiple reviews [16,17] and a recent report [7] using a decision tree analysis from 48 LMIC countries reported that gender-related factors play a strong role in perinatal mental health problems such as anxiety and depression, including in the South Asian context [18].

A few interventions for perinatal health in general that have addressed gender-related factors and focused on enhancing support systems for pregnant women or addressing gender inequity in the family have found improvement in rates of postpartum depression as well [19,20,21]. When specifically looking at interventions, literature suggests how addressing gender gaps with regards to perinatal health programs and policies can be an active agent of change in addressing low mobility, female genital mutilation, unintended pregnancy and increased preference for a male child [22].

Yet, different research reviews related to gender-based violence, and family and reproductive health have notably overlooked the practical ways of overcoming gender inequality to improve perinatal mental health [23]. Gender-based interventions in perinatal health may be gender intentional, gender accommodative or gender transformative. Gender intentional means identifying and understanding gender inequalities [24]. Gender-accommodating interventions seek to compensate for gender norms and ideally decrease *existing* inequalities; however, they are not aimed at role-reversal or changing gender norms [25]. Gender transformative interventions, on the other hand, are interventions that create opportunities for individuals to actively challenge gender norms and address power inequities between persons of different genders [26]. Principles of Gender Transformative Approach (GTA) go beyond improving healthcare systems and access for women alone but also include men, children and other family members to promote better health for communities, as a whole. However, while risk factors have been studied, gender-accommodative or transformative interventions have not been designed from a curative or a preventative lens [27]. Fathers have been involved at best as only one part of an intervention and these too have not addressed gender transformation [28,29,30]. Since gender transformative interventions aid in bridging the gender gap [26,27,31,32], it is necessary to develop interventions for perinatal mental health with a Gender Transformative (GT) lens, especially in countries with a large gender equity gap. This is especially true for the more prevalent perinatal mental health concerns such as depression and anxiety which are driven by a host of psychological and social risk factors.

Presently, there is meagre evidence based data on perinatal mental health interventions that are gender-transformative in LMIC countries. To examine available evidence as well as highlight the various components of the intervention and to encourage research in this area, this review attempts to (a) review the literature on GT interventions that have been implemented in LMIC, and have addressed perinatal mental health directly or indirectly, (b) describe components of these interventions and (c) based on this, provide a framework of action and recommendations for designing gender transformative intervention for perinatal mental health care, specifically for early identification and treatment of perinatal anxiety and depression in the community.

## 2. Method

This scoping review was conducted by utilising the framework developed by Arksey and O’ Malley’s [33] that included five stages, namely (1) identifying the research question that was broad in nature, (2) identifying relevant studies, a process that remains comprehensive and strategic, (3) select studies based on inclusion/exclusion criteria, based on familiarity with literature, (4) charting data related to key themes and issues and lastly, (5) collating, reporting and summarising the results which could be descriptive, thematic and/or numerical in nature [34]. The findings are reported according to the Preferred Reporting Items for Systematic Reviews and Meta-Analyses extension for Scoping Reviews (PRISMA-ScR) checklist [35]. The sixth stage of a scoping review—stakeholder consultation—is an optional element and was not included in this review.

### 2.1. Identifying the Research Question

The research question for this scoping review was “what is known about gender transformative interventions that address perinatal mental health in Low and Middle Income Countries?” The specific research questions for the present reviews were as follows: (1) What are the objectives and purpose of developing gender transformative interventions for perinatal mental health in the context of the LMICs? (2) What are the components of these interventions (3) What are the kinds and contexts of male-involvement/engagement in these gender-transformative perinatal mental health interventions? and (4) What are the outcomes for perinatal mental health following these interventions?

### 2.2. Identifying Relevant Studies

The following electronic databases were searched for English language publications between 2012 and present: PubMed, PsycInfo, Scopus, Web of Science. Journals were inclusive of Sage Publications, SpringerLink, Taylor and Francis, Wiley Online, and Oxford University Press. Other databases searched include Google Scholar. The authors also manually searched the reference lists of included papers, reports and other reviews to identify further eligible papers or studies.

A comprehensive search using keywords included Gender transformative interventions LMIC; Gender transformative interventions for maternal healthcare; Gender transformative interventions for perinatal healthcare and Gender transformative interventions for perinatal and maternal mental health. The specific search terms for studies included Perinatal OR mental health outcomes OR antenatal OR pregnancy OR childbirth OR postpartum OR postpartum depression OR maternal OR perinatal partner support OR involvement of fathers OR mental health outcomes OR perinatal IPV OR perinatal GBV [in title, abstract, keywords] AND male engagement interventions OR family interventions AND LMIC/Low and Middle income countries (in all).

### 2.3. Study Selection (Inclusion and Exclusion)

The inclusion and exclusion criteria were developed by the authors and are shown in Box 1. The search period ranged from studies published between the years of 2012–present. A longer time period was utilised to map the scope of literature owing to the limited number of studies that focused on gender transformative interventions that specifically focused on maternal or perinatal mental health and wellbeing. Moreover, to ensure a rigorous search, data from grey literature were also included. A quality assessment of the studies selected was carried out by two senior researchers (PS, VS). Studies that developed gender transformative interventions inclusive of a component that targeted males or extended family members’ involvement were reviewed.

Additionally, systematic reviews of interventions that targeted prevention or reduction of violence against women, girls or mothers were also included in the review. Since maternal and child’s health and wellbeing are strong predictors of mental health [36,37], psychosocial interventions that were gender-transformative and addressed maternal health and care were also included. Inclusion criteria consisted of studies conducted in the LMIC region; countries belonging to LMIC were classified according to the Society for the Study of Human Biology [38]. For the current 2022 fiscal year, low-income economies are defined as those with a GNI per capita, calculated using the World Bank Atlas method, of USD 1045 or less in 2020 (World Bank Country and Lending Groups, 2020) [36]. The following exclusion criteria were applied: study location not in LMIC, not relating to perinatal period, not relating to mental health and not relating to maternal mental health.

Box 1Inclusion and Exclusion criteria.Inclusion criteriaPublished in EnglishPrograms designed and implemented in LMIC countries.Studies having any one of the three mental health outcomes in the perinatal period which include
a.Improvement in social support or better relationship with partner;b.Decrease in depression, anxiety or any Common Mental Disorder;c.Decrease in Domestic violence or Intimate Partner Violence.Any study design—RCTs, Non Randomised Controlled studies, Case Control studies, pre post intervention studies.Publication years 2007–2022.Exclusion criteriaStudy location not in LMIC.Studies not relating to perinatal period.Studies not relating to mental health or risk.Studies not relating to maternal mental health.

### 2.4. Charting the Data

Data were extracted according to the PRISMA-ScR [37,39] checklist [35]. The primary objectives, study characteristics (author, year, country/region and outcome measures), study population, components of interventions, primary outcomes and aspects of male engagement were tabulated in line with the research questions.

### 2.5. Reporting the Results

Using a narrative approach, these interventions were critically analysed by VS, AR and PC and reported in terms of intervention characteristics, risk of bias/methodological quality, categorisation of outcomes and identification of gaps in evidence as previously noted in scoping reviews that focused on GT interventions [40]. A total of 6 studies were then finalised and analysed descriptively to understand GT in perinatal care and GT in the context of LMIC.

## 3. Results

From the search process, 16 studies were identified since they met the inclusion criteria. However, 10 studies were excluded because 7 studies focused on gender but not on the gender “transformative” aspect and did not assess mental health outcomes and 3 studies did not assess perinatal mental health conditions. Figure 1 illustrates the PRISMA flowchart of our search strategy. Table 1 highlights the data summary obtained from the 6 studies and additionally displays the study characteristics, study population, components of interventions, primary outcomes and aspects of male engagement.

### 3.1. Article Characteristics

The six finalised studies ranged primarily between 2015–2022, although the study attempted to include studies over a period of 15 years (refer to Table 1). Only one study was conducted in 2008 [41]. The results were further indicative of how there were very few studies that focused on understanding perinatal mental health by implementing gender transformative interventions. All the six studies included men and women/husbands and wives. One study focused on women and husbands as well as extended family members [41]. All the interventions included a male engagement component in the intervention. All the interventions were mostly conducted in rural or semi-urban areas.

Most of the identified studies utilised group discussions as the basis to facilitate critical dialogue and awareness regarding gender roles and mental health. In the Bandebereho couples’ intervention, from Rwanda, small groups of critical reflection and dialogues were initiated with couples and men [44]. Similarly, the Counseling Husbands to Achieve Reproductive Health and Marital Equity (CHARM) intervention implemented in Maharashtra, India, also involved sessions based on gender and culture for individuals to explore how gender roles influenced wellbeing for mothers [23]. Two studies in Zimbabwe and Congo [42,45] addressed how developing positive models of masculinity can decrease gender disparity by initiating men’s groups and engaging in group discussions. Study designs ranged from randomised controlled trials [23,42,44,45] to pre and post evaluation designs [23,44]. In the following sections studies are presented thematically based on their objectives and primary outcomes. Overall, the studies highlighted the need to address perinatal mental health concerns through the involvement of other family members or husbands through a gender-transformative lens.

### 3.2. Program Evaluation

This section reports practical aspects of the interventions (Table 2) and important components of GT interventions that were designed, namely, to decrease gender-based violence, improve maternal mental health, and improve couples’ relationships.

A successful couples’ intervention designed by Doyle et al. [44], implemented in Rwanda, focused on engaging men and their partners in a participatory group session consisting of critical reflection and dialogue. Addressing power relations in the community demonstrated substantial improvements with respect to marriages and modern contraceptive uses. Therefore, while it did not address perinatal healthcare directly, the couple-focused interventions had longer term implications with respect to perinatal and maternal mental health [47]. The intervention induced a significant positive impact on maternal health by reducing instances of physical and sexual IPV. It also increased male accompaniment to antenatal care and decreased dominance of men in decision-making challenging existing gender norms. Components of intervention involved training community volunteers (local fathers) to co-facilitate sessions on pregnancy, family planning and marital communication. Sessions involved ice-breakers, group activities, games and media such as cartoons and short films [48] However, the research was implemented only for 12 months, leading to an unsustainable effect. This explains why despite greater male involvement, women’s time spent on labour at home remained the same. Moreover, since behavioural changes were self-reported there is a risk of participants (both men and women) providing desirable answers.

A similar and older study from India targeted young married women, their husbands as well as family members to address and modify gender norms [41]. The study attempted to address communication and decision-making in the family by empowering women and creating supportive social structures by providing interventions to husbands and mother-in-laws through the First Time Parent Project in rural West-Bengal and Gujarat. Components of intervention included education and counselling sessions for young married women, training, outreach programs and workshops for husbands and mothers-in-law as well as developing support groups for women. While the study did not address maternal mental health directly, it addressed risk factors for poor maternal mental health since primary outcomes involved decreased gender-based violence and improving support in homecare practices. The study did not however ensure follow-ups and therefore, the effectiveness of the intervention remains unclear. The study however provided insightful recommendations that included allying influential members of a family within the interventions as well as creating a support network for different groups of mothers, such as those trying to conceive, delay the first pregnancy and new mothers.

Raj et al. [23] conducted a randomised controlled trial evaluation in India, across 50 geographic clusters in rural Maharashtra, which primarily focused on gender equity and family planning for men and couples. Based on the baseline scores of contraceptive behaviours and IPV attitudes, a CHARM intervention, comprising three sessions of family planning, gender-equity for couples was utilised. Sessions involved discussing gender-equity through pictorial flipcharts that addressed family planning, barriers to family planning and respectful marital communication. Results indicated that contraceptive communication increased and decreased intimate partner violence amongst couples was reported at an 18-month follow-up. While GT interventions have shown their efficacy across countries, in LMIC, since resources are low, it is imperative to note that investment interventions such as home visiting programs need to be maximised to prevent IPV or child maltreatment [48].

Another intervention that was targeted for women, children and men/fathers/co-parents in Mutasa district, Zimbabwe, used community-based training and discussion groups that addressed services for mothers, HIV transmissions and engaging in problem-solving therapy. Results indicated that addressing gender inequality improved maternal mental health [46]. The interesting aspect of this program included creating and implementing educational and outreach programs that encouraged “male champions”. Separate tools were prepared for men and women participants. Interventions for women were delivered by local female village health workers through Participatory and Learning Action (PLA) cycles. For men, interventions involved male project members discussing gendered-division of labour, safe sex and men’s contribution in care-seeking behaviour. The intervention successfully integrated gender equality and male engagement, leading to increased couples communication, reduced maternal workload and increased nutrition during pregnancy, another paramount implication included increased value of girl children.

Along the lines of men’s involvement, the study by Bapolisi et al. [45] focused on investigating the impact of men’s involvement on women’s health and child nutrition. The primary focus in this study was to engage men for more gender equality, expecting a positive effect of this combined intervention on the household economy, on child nutritional status, on the use of reproductive health services including family planning, and on reducing sexual and gender-based violence (SGBV). The intervention in the aforementioned study involved developing positive masculinity by engaging men using a peer-based approach. Reflective conversations were conducted through Gender-Dialogue Groups (GDGs) facilitated by both one male and one female, trained as gender-based violence field agent and economic recovery field agent. Men were encouraged to adopt attitudes and behaviours that promoted women’s economic empowerment as well as reduced gender-based violence. The study provided insightful implications regarding gender-power dynamics on both household as well as community levels. It can further be noted that through participatory interventions, mental health services inclusive of antenatal care, maternity and family planning can be improved through male-involvement.

A recent study that contributed towards the current limited literature, was Comrie-Thomson et al.’s [46] trial on implementing a gender-synchronised intervention. Gender-synchronised interventions are conceptualised as programs that employ multiple strategies to change community norms related to gender as well as engage men to achieve gender equality and improve health [49]. As a part of the intervention, women participated in PLA cycles conducted through monthly one hour group discussions, facilitated by female village health workers in a central community location. Men, on the other hand, participated in monthly one-hour group discussions, facilitated by the male project staff member in men’s workplaces or a central community location. Group discussions rooted in problem-solving therapy, focused on topics such as home care practices during pregnancy along with various gender-related challenges women faced. Results indicated that women reported decreased postnatal depression scores and care-seeking as well as relationships significantly improved.

However, the aforementioned GT interventions particularly focused on prenatal, maternal care and personal empowerment. Alternatively, it is indicative of how programs and interventions should focus more intentionally on postnatal mental health care, particularly gender-intentional postpartum family planning interventions to ensure antenatal and intrapartum care that were earlier provided remained sustainable and effective. By implementing these interventions, it can be inferred that perinatal mental health will significantly improve eventually decreasing mental health concerns such as anxiety and depression. Possible adverse events that have to be considered includes increased tension in parent relationships and familial relationships due to changed expectations and behaviours [50]. Therefore, developing GT interventions to explicitly address power dynamics, values and norms throughout perinatal, prenatal and postnatal maternal health as well as mental health remain a necessity to improve quality of care sustainably.

## 4. Discussion

Due to limited studies, the review could not identify major contexts in which GT interventions were designed. However, it was vastly noted that GT interventions had multifold implications with respect to improvement on mental health, maternal mental health, decreased IPV and GBV and improved couples’ relationships and homecare practices. It was previously noted that gender related factors seem to play a strong role in mental health problems in pregnancy and the postpartum especially in the LMIC region [51]. Therefore, this review specifically identified key program components that may have contributed to positive mental health outcomes as well as improvement in social and partner support and decrease in IPV or DV, factors that have a strong link with depression and anxiety in the perinatal period.

Similar to previous reviews and studies, the results in the present paper indicate how main components of interventions to improve maternal mental health consist of quality time with the infant, group sessions with husbands or family members, counselling sessions and psychoeducational sessions [2,52]. Interventions are further indicative of how addressing gender disparities can significantly lead to positive outcomes. Moreover, apart from addressing maternal care specific to infant care, it also becomes necessary to address the general wellbeing of mothers through empowering GT interventions. These results demonstrate how development of gender transformative interventions is necessary to improve mental health outcomes long term, amongst mothers [53].

In programs that focused on fatherhood, targeting men, fathers and husbands, interventions mostly focused on educational sessions on gender-roles [44,54] indicating that addressing gender roles and norms was an important component of GT interventions as highlighted by studies previously [30,51]. Recent studies have highlighted the necessity of male engagement in maternal and perinatal healthcare [25,44]. Moreover, findings accordingly highlighted that GT interventions which focused on fatherhood helped in transforming harmful masculine norms which underpin gender-based violence. While these interventions aim for men to increase their involvement in their partner’s pregnancies and accompany them to health services, it is necessary to note that increasing male engagement as a strategy should include ethical considerations to ensure men do not assume the stance of “protecting” and “looking after” women which in turn can cause power imbalance and gender disparities.

Furthermore, fatherhood programs when designed from a systemic lens can significantly support and protect women, families and children from violence. In support, a systematic review of male engagement in GT interventions for women in the community highlighted that 11 out of 12 GT interventions revealed a significant change in men’s attitudes towards gender norms, establishing gender equality [31]. This is suggestive of the need to develop and design more GT interventions that focus increasingly on the aspect of male engagement. However, this study did not focus on women in the perinatal period.

Further, the aforementioned studies do not acknowledge the structural norms that influence masculinity and how norms related to masculinity are also changing [31]. Moreover, while the results are indicative of group education, community outreach and mass-media campaigns are all effective program interventions, none of the studies focus on evaluating long-term change. It remains unclear how family members, caregivers, and men will continue to succeed in sustaining their short-term change in the absence of contextual and structural changes.

To achieve long-term, sustainable change, community-level interventions need to be accompanied by policies that support the changes men undergo through GT interventions. Future recommendations for gender transformative interventions include taking a more relational perspective that attempts to integrate men and boys with efforts to empower women without adapting the attitude of “saving women and families” since it was noted that many men engaging in activism for equality or trials that promote equality are not disconnected from an inherent saviour complex [51].

Studies could also focus on developing strategies that address change at the level of families and communities leading to sustainable changes through involvement of other family members, which also result in sustainable and lasting effects. Additionally, future studies could focus on programmatic efforts on gender barriers that accompany life stages, that range from helping newly married couples with no children to delay in their first pregnancy. Developing GT interventions for specific groups such as those facing perinatal loss and adolescent mothers is needed. GT studies need to provide more information about how men were encouraged to participate in these trials and stay engaged through the course of multiple sessions, which components were preferred by the groups and whether the gender of the facilitators made a difference and if groups should be men only or combined. This information will enable future researchers to help in better planning of future GT intervention studies for perinatal mental health including deciding the “dose” of the intervention in addition to the methods.

Moreover, some reports [55,56] focusing on male engagement in perinatal mental health, from High Income Countries, have used technology such as SMS and other online tools. Countries in the LMIC region also reported using technology to improve perinatal mental health [57,58,59]; however, the interventions did not include a male engagement component. It is possible that technology-based interventions can increase accessibility as well as be useful in involving difficult-to-engage men.

## 5. Limitations

It is necessary to remain cautious while interpreting the findings of this review because the evidence is limited due to the small number of studies that were identified. Moreover, since the synthesised studies were not methodologically similar, the findings of the current review cannot be generalised. Our findings and recommendations are partially informed by the studies and their limitations, also accounted for, in our analysis. Moreover, given the sensitivity and stigmatised nature of these issues, consideration must be given to the presence of social desirability bias which may have influenced disclosure and involvement of participants in the programs. Future research may examine the dose or the optimum length of the intervention itself and on whether treatment gains are maintained over longer follow-up periods.

## 6. Conclusions

Our review has highlighted a need for GT interventions that focus on male engagement, family members and community as a whole. Our review also highlights the methodological strengths and deficits in existing interventions, paving the way for future research to address these limitations and mindfully develop programs that yield effective maternal health outcomes. Lastly, by emphasising programs implemented in the LMIC region, our review addresses the need to develop shared goals that address gender-based violence, cultural norms and family dynamics since perinatal and maternal mental health outcomes cannot be improved in isolation.

## Figures and Tables

**Figure 1 ijerph-19-12357-f001:**
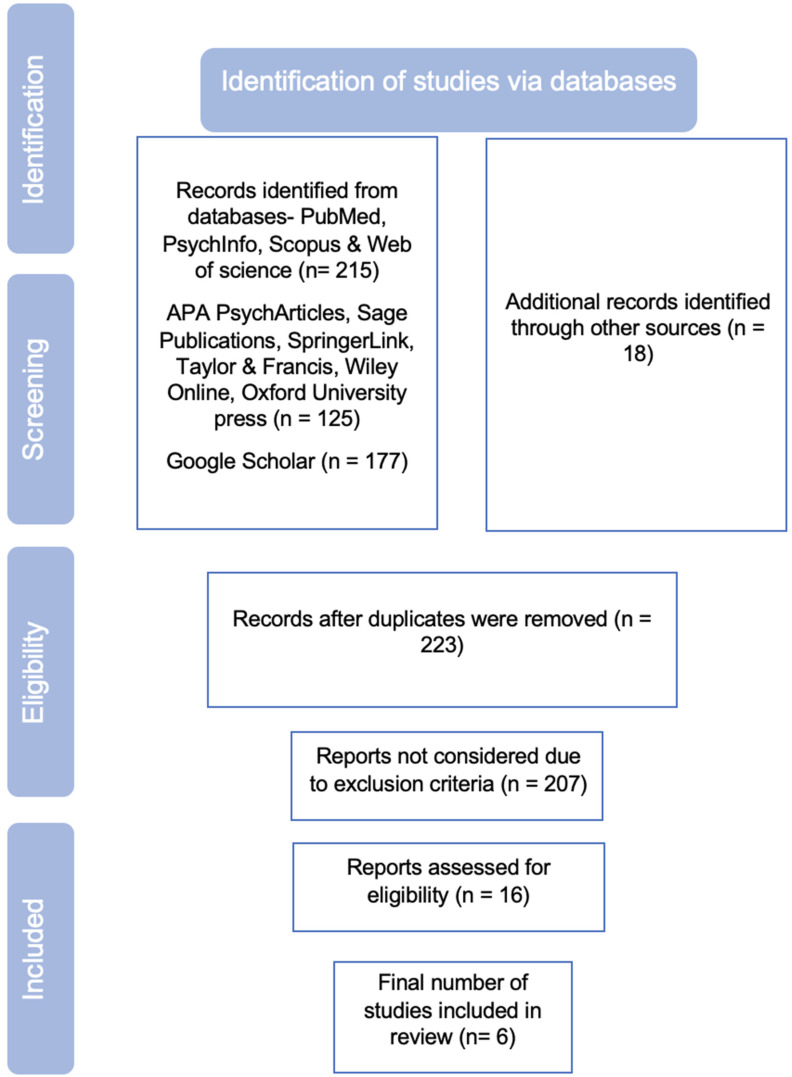
PRISMA flowchart of the search strategy.

**Table 1 ijerph-19-12357-t001:** Description of study population and components of intervention.

S/No	Author, Year and Country	Target Population and Sample Size	Gender-Transformative Components	Aspect of Male Engagement	Primary Outcomes (With Data if Possible)
1	Santhya et al., 2008 [41] West Bengal and Gujarat, India	Women (pregnant and post-partum first time mothers), husbands and family members 2115 women	The First Time Parent project targeted young married women and their husbands as well as family members to modify gender norms and support prenatal as well as maternal healthcare behaviours.	Outreach workers interacted with husbands about pregnancy and delivery plans. Husbands received home visits from male outreach workers.	Decision-making increased for married women in their household (61%). More women adhered to egalitarian gender attitudes (38%) in Diamond Harbour; however, no difference was observed in Vadodara. Positive change in women’s perception about wife-beating and domestic violence (42%). No significant differences between age and religion was found between the intervention and control group.
2	Comrie-Thomson, L. et al. (2015) [42] Bangladesh, Tanzania and Zimbabwe	Married men and their wives 237 males and females	Education and outreach were conducted with men’s groups and individual men through designated gender equality champions, peer educators or role models. Dialogue, education and mobilisation were conducted with traditional and religious leaders, who have influence over community beliefs and behaviours. Integrated gender equality and male engagement messages delivered through a wide range of activities including community Theatre for Development (T4D), community radio (in Barguna), and community meetings.	Education and outreach were conducted with men’s groups and individual men through designated gender equality champions, peer educators or role models.	Male and female participants identified many benefits associated with male engagement in MNCH, including improved health outcomes for women, newborns and children. Increased couple communication and improved couple relationships. Increased maternal nutrition and rest during pregnancy. Increased value of girl children. Increased assistance of fathers in household chores (41.7%). Assisting wives to access healthcare services (57.7%). Increased couple communication and shared decision making.
3	Raj et al., 2016 [23] Yore et al., 2016 [43] Maharashtra, India	Married men and their wives 1081 Rural young husbands and their wives	The intervention involved three gender, culture and contextually-tailored family planning and gender equity (FP + GE) counseling sessions. A desk-sized CHARM flipchart was used by village health providers to provide men and couples with pictorial information on family planning options, barriers to family planning use including gender equity-related issues (e.g., son preference), the importance of healthy and shared family planning decision-making, and how to engage in respectful marital communication and interactions (inclusive of no spousal violence in the men’s sessions).	Counseling Husbands to Achieve Reproductive Health and Marital Equity (CHARM) intervention, a multi-session intervention delivered to males alone, but included a session with their wives.	Findings document that women from the CHARM condition, relative to controls, reported increased contraceptive use at 9-month follow-up (55.7%). They were less likely to report physical IPV at 18-month follow-up (48%). Men in the CHARM condition were less likely. to report attitudes accepting of sexual IPV (51%). No significant time by treatment effects were seen for sexual IPV between the control and intervention group.
4	Doyle et al., 2018 [44] Rwanda	Expectant/current fathers and their partners (pregnant women) 575 couples and 1123 men	The Bandebereho couples’ intervention engaged men and their partners in participatory, small group sessions of critical reflection and dialogue. In Rwanda, the MenCare+ program was known as Bandebereho, or “role model”, as it aimed to transform norms around masculinity by demonstrating positive models of fatherhood.	Transform norms around masculinity by demonstrating positive models of fatherhood. Sessions addressed: gender and power; fatherhood; couple communication and decision-making; IPV; caregiving; child development; and male engagement in reproductive and maternal health.	Compared to the control group, pregnant women in the intervention group reported: less past-year physical and sexual IPV, greater attendance and male accompaniment at antenatal care (61.17%). Pregnant women (79.15%) and men (57.71%) in the intervention group reported: less child physical punishment. Women reported greater modern contraceptive use and less dominance of men in decision-making (56.08%). However men’s level of participation in childcare between the intervention group and control group remained the same.
5	Bapolisi et al., 2020 [45] Democratic Republic of Congo	Husbands and wives 800 men and women	The “Mawe tatu” program, links Village Savings and Loans Associations (VSLA) for women with men-to-men sensitisation to transform gender-inequitable norms and behaviours for the empowerment of women. Comprehensive sexuality education for young people, which includes gender and rights themes, is offered as well.	Developing “positive masculinity” by engaging men, if possible spouses of VSLA’s members, towards women’s rights using a peer-to-peer approach.	The primary outcomes are to engage men for more gender equality, expecting a positive effect of this combined intervention on the household economy, on child nutritional status, on the use of reproductive health services including family planning, and on reducing sexual and gender-based violence (SGBV). Note: Data on the study are not yet published.
6	Comrie-Thomson et al., 2022 [46] Manicaland, Zimbabwe	Women and male co-parents 433 women (Pregnant and post-partum mothers up to 2 years post-pregnancy) and 273 men	Women participated in Participatory learning action (PLA) cycles conducted through monthly one-hour group discussions. Discussions explored MNCH services and home care practices recommended during pregnancy and between zero and two years of age, including services for the prevention of mother-to-child transmission of HIV (PMTCT). The +Men component was delivered by a trained male OPHID staff member who was also a nurse and midwife with substantial community development experience, targeting men. Men participated in monthly one-hour group discussions. Discussions explored similar health topics to those addressed in women’s groups and the same flip chart was used to present information.	Men participated in monthly one-hour group discussions, facilitated by the male project staff member in men’s workplaces or a central community location.	Primary outcomes of interest reported Decreased symptoms of depression and anxiety (63%). Increased women’s participation in decision-making (68.7%). Improved men’s gender attitudes, and couple relationship dynamics (88.7%). Increased practical support provided by men (78.4%). No effect was detected on the proportion of men participating in antenatal care consultations, supporting childbirth by providing money or goods, contributing to household chores during pregnancy or after childbirth, encouraging their pregnant coparent to rest, or settling their baby at night.

**Table 2 ijerph-19-12357-t002:** Details of the facilitators and recipients of the intervention.

S/No	Study Design (Format)	Male Engagement Intervention	Individual/Couple/Group Intervention	Facilitators	Inclusion of Other Family Members
1	Randomised controlled trial	MenCare+ program	Couple based intervention. Men along with their current partners (pregnant wonen) were included. Men alone were invited for 15 sessions and with their partners were invited for 8 sessions.	Sex-matched interviewers from Laterite, who had no involvement in the intervention, conducted the interviews. Community volunteers (local fathers) met with the same group of 12 men/couples on a weekly basis. The volunteers received a two-week training, material support, and refresher training. Local nurses and police officers co-facilitated the sessions on pregnancy, family planning, and local laws, respectively.	No
2	Cluster randomised controlled trial	CHARM gender-equity (GE) counselling in family planning (FP) services. The intervention involved three gender, culture and contextually-tailored family planning and gender equity (FP + GE) counselling sessions delivered by trained male village health care providers to married men (sessions 1 and 2) and couples (session 3) in a clinical setting.	Couple-based intervention	CHARM providers were allopathic (n = 9) and non-allopathic (n = 13) village health care providers trained over three days on FP counselling, GE and IPV issues, and CHARM implementation. All VHPs in the study villages were male; 22 VHPs were trained for delivery.	No
3	Qualitative study	Focus-group discussions and in-depth interviews	Men-only counselling Couples counselling	Male community workers engaged in men-only education group sessions.	No
4	Cluster-randomised, longitudinal intervention study	Positive masculinity groups	Only men peer-to-peer discussion groups	Information not given	No
5	A cluster-randomised controlled pragmatic trial	+Men component	Only men discussion groups Men and women were assessed separately for baseline scores.	Trained male Public Health Interventions and Development (OPHID) staff members.	No
6	A quasi-experimental research design	First-time Parents Project	Women-only sessions from female outreach workers. Husbands received home-visits from male outreach workers.	Same-sex facilitators conducted interventions.	Yes. Mothers and mother-in-law were included for home-visit based interventions (family sessions).

## Data Availability

Not applicable.

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
