# Peer review of "Gender Transformative Interventions for Perinatal Mental Health in Low and Middle Income Countries—A Scoping Review"

_ijerph, 2022, doi:10.3390/ijerph191912357_

Round 1
Reviewer 1 Report
Congratulations on a great review article. I understand the difficulty in making this article based on the scarcity of well-designed studies. Would it be possible to give several recommendations for the upcoming studies in the discussion/conclusion section? Such as the need to follow up after intervention or the length of study.
Author Response
Thank you for this comment. We have addressed this point in the discussion section.
Reviewer 2 Report
This is a review paper gathering the gender-transformative (GT) interventions from Low-and-Middle Income Countries (LMIC) and trying to identify components of the useful and acceptable interventions. To be honest, it’s my first time reviewing a paper not mainly based on data but on intervention content. The authors wrote the paper in a good logic, provided detailed results and gave insights in discussion section. I have some minor comments and hopefully they can help improve the manuscript.
1. Though there is a paragraph in the discussion section about the strategy from High Income Countries (page 12), as the world is developing and new technologies come out quickly, it can be interesting for the audience if the authors could discuss how the LMIC can learn from HIC more efficiently. In addition, it would be great if the authors can add some discussion about what are the components of GT interventions and their differences from findings in this paper.
2. I understand the authors provided the criterion for LMIC (row 136), but it’s important to know how many countries are included and the distribution of these countries around the world.
3. Figure 1: please consider adding arrows into the flow chart. What’s the exclusion criteria between the last two boxes “n=16” and “n=6”?
4. Table 1: please consider simplifying the content in table 1 by using key words instead of sentences in the table cells.
5. Please provide the full names of abbreviations at their first appearance in the manuscript. For example, on rows 125 and 147, did the VSs represent the same or not?
6. Please carefully read through the manuscript and revise the spelling or grammar errors. For example, on row 86 ‘select’ to ‘selecting’; please search ‘focusof’ on page 10 and ‘future programs’ on page 11. Moreover, line numbers are missing from page 9.

Author Response
Thank you for the useful comments. Please find our responses below.
Though there is a paragraph in the discussion section about the strategy from High Income Countries (page 12), as the world is developing and new technologies come out quickly, it can be interesting for the audience if the authors could discuss how the LMIC can learn from HIC more efficiently. In addition, it would be great if the authors can add some discussion about what are the components of GT interventions and their differences from findings in this paper.
We have addressed this in the concluding paragraph of the discussion and the conclusions.
I understand the authors provided the criterion for LMIC (row 136), but it’s important to know how many countries are included and the distribution of these countries around the world.
We have used a standard accepted definition for LMIC and have a reference for any further information.
Figure 1: please consider adding arrows into the flow chart. What’s the exclusion criteria between the last two boxes “n=16” and “n=6”?
We have clarified this.
Table 1: please consider simplifying the content in table 1 by using key words instead of sentences in the table cells.
This is a descriptive table. Hence, relevant details about the included studies have been succinctly presented.
Please provide the full names of abbreviations at their first appearance in the manuscript. For example, on rows 125 and 147, did the VSs represent the same or not?
They are the same author.
Please carefully read through the manuscript and revise the spelling or grammar errors. For example, on row 86 ‘select’ to ‘selecting’; please search ‘focus of’ on page 10 and ‘future programs’ on page 11. Moreover, line numbers are missing from page 9.
These have now been addressed.